# The Battle between Bacteria and Bacteriophages: A Conundrum to Their Immune System

**DOI:** 10.3390/antibiotics12020381

**Published:** 2023-02-13

**Authors:** Addisu D. Teklemariam, Rashad R. Al-Hindi, Ishtiaq Qadri, Mona G. Alharbi, Wafaa S. Ramadan, Jumaa Ayubu, Ahmed M. Al-Hejin, Raghad F. Hakim, Fanar F. Hakim, Rahad F. Hakim, Loojen I. Alseraihi, Turki Alamri, Steve Harakeh

**Affiliations:** 1Department of Biological Sciences, Faculty of Science, King Abdulaziz University, Jeddah 21589, Saudi Arabia; ateklemariam@stu.kau.edu.sa (A.D.T.); rhindi@kau.edu.sa (R.R.A.-H.); mgalharbi@kau.edu.sa (M.G.A.); jkiravu@stu.kau.edu.sa (J.A.); aalhejin@kau.edu.sa (A.M.A.-H.); 2Department of Anatomy, Faculty of Medicine (FM), King Abdulaziz University, Jeddah 21589, Saudi Arabia; wsaadeldin@hotmail.com; 3Department of Anatomy, Faculty of Medicine, Ain Shams University, Cairo 11566, Egypt; 4Microbiology Level 2 Laboratory, King Fahd Medical Research Center, King Abdulaziz University, P.O. Box 80216, Jeddah 21589, Saudi Arabia; 5Ministry of Health, Jeddah 11176, Saudi Arabia; raghad_hakim@hotmail.com; 6Department of Internal Medicine, King Abdulaziz University, Jeddah 21589, Saudi Arabia; fanar.f.hakim@gmail.com; 7Ibn Sina National College for Medical Studies, Jeddah 21418, Saudi Arabia; rehad_21@hotmail.com (R.F.H.); lojeenibrahim@icloud.com (L.I.A.); 8Family and Community Medicine Department, Faculty of Medicine in Rabigh, King Abdulaziz University, Jeddah 21589, Saudi Arabia; olbalamri7@kau.edu.sa; 9King Fahd Medical Research Center, Yousef Abdullatif Jameel Chair of Prophetic Medicine Application, Faculty of Medicine, King Abdulaziz University, Jeddah 21589, Saudi Arabia

**Keywords:** bacteriophage, phage resistance, phage counterstrategies, prophage, phage therapy

## Abstract

Bacteria and their predators, bacteriophages, or phages are continuously engaged in an arms race for their survival using various defense strategies. Several studies indicated that the bacterial immune arsenal towards phage is quite diverse and uses different components of the host machinery. Most studied antiphage systems are associated with phages, whose genomic matter is double-stranded-DNA. These defense mechanisms are mainly related to either the host or phage-derived proteins and other associated structures and biomolecules. Some of these strategies include DNA restriction-modification (R-M), spontaneous mutations, blocking of phage receptors, production of competitive inhibitors and extracellular matrix which prevent the entry of phage DNA into the host cytoplasm, assembly interference, abortive infection, toxin–antitoxin systems, bacterial retrons, and secondary metabolite-based replication interference. On the contrary, phages develop anti-phage resistance defense mechanisms in consortium with each of these bacterial phage resistance strategies with small fitness cost. These mechanisms allow phages to undergo their replication safely inside their bacterial host’s cytoplasm and be able to produce viable, competent, and immunologically endured progeny virions for the next generation. In this review, we highlight the major bacterial defense systems developed against their predators and some of the phage counterstrategies and suggest potential research directions.

## 1. Introduction

Antimicrobial resistance is a major global public health concern and approximately 10 million people will die yearly worldwide by 2050 because of antimicrobial resistance [1]. Consequently, novel therapeutic strategies have been sought and must be developed and used as an alternative to antibiotics or in conjunction with conventional therapy. One of these approaches involves the use of bacteriophages (phages). Phages are the most abundant predators of bacteria in nature. According to viral ecologists, phage infections happen about 10^23^ times per second worldwide, indicating a very dynamic and large population [2]. A significant portion of the genes (over 80%) encoded by phages have neither been linked to known proteins nor have their functions been identified. This makes them the greatest source of genetic novelty in the biological world [3].

Bacterial infections that are difficult to treat can be either treated with phages alone or in conjunction with antibiotics [4]. Currently, phage therapy is being reevaluated as a potential alternative to classical therapeutic agents in the Western countries. However, there are still multiple challenges to overcome. These challenges include phage resistance risks, immunity to phages, problems associated with appropriate phage selection, and new regulatory requirements [5].

Bacteria and phages are seemingly involved in a continuous battle. It is part of continuous cycles of coexistence and evolution, resulting in phage-resistant hosts protecting bacterial lineages, while counter-resistant phages threaten such strains. Phages, by developing resistance, play a crucial role in controlling bacterial populations in most, if not in all, the milieus. Bacteria can evade the phage attack via several mechanisms and some of these strategies include the following: DNA restriction-modification (R-M), spontaneous mutations, blocking of phage receptors, production of competitive inhibitors, and extracellular matrix and acquired immunity via the clustered regularly interspaced short palindromic repeats and associated proteins (CRISPR-Cas) mechanism [6,7,8]. On the contrary, phages developed several counterstrategies and circumvent the phage resistance warfare. These host-phage interaction is a complex and multifaceted process, which influences the diversity of genetic makeup of both bacteria and their predators, and it is one of the driving forces creating genetically fit populations from both sides [9].

In this review, we present an overview of the major anti-phage defense strategies of bacteria and the counterstrategies used by phages to evade these systems and suggest potential research directions.

## 2. Mechanisms of Phage Resistances

The mechanism of phage resistance could take place at various stages of phage replicative cycle. During the phage replication cycle, a phage introduces its DNA via translocation into the host’s cytoplasm. This will lead to either the lysogenic cycle (prophage formation) or the lytic cycle. In the lytic cycle, phages pass via several steps and early and late gene expression, which leads to maturation and aggregation of newly produced virion, which are ultimately released via lysis. The phage resistance strategies can interfere with one of these steps as shows in Figure 1 [8], and each resistance mechanism is described in detail below.

### 2.1. Preventing Phage Adsorption

Phage adsorption to the bacterial cell surface is performed through specific receptors as the first step of phage infection cycle. Several bacterial cell surface proteins, lipopolysaccharides, and other surface polysaccharide and carbohydrate moieties can serve as receptors for phages [10]. Bacteria have generated several barriers to prevent phage attachment to their cell surface, such as hidden receptors with extracellular matrix [11]. Bacterial extracellular matrix (a loose network of polymers), such as slime layers (e.g., *Campylobacter fetus* slime layer) or capsules (e.g., *Escherichia coli* K1 capsule, and *Klebsiella pneumoniae* capsule) can cover the bacterial surface and makes the phage receptors inaccessible for phage binding so that protecting bacteria from phage attack [12]. Some bacterial strains can synthesize competitive inhibitors and mutate (modify) their receptor or by generating receptors, which are new for the virus [13] (Figure 2). The phage receptor diversity on the surface of the host cell are affected by phage co-evolutionary adaptations to stun these barriers [10]. This includes phase variation and diversity-generating retro elements (DGRs) [10]. Phase variation is a reversible, as well as heritable process, regulating the expression of bacterial gene, thereby genes can shift between a non-functional state and a functional existence ensuing to phenotypic variations among the bacterial community even among strains which have similar genotype. Gencay et al. studied the mechanism of resistance developed by *Campylobacter jejuni* (strain NCTC11168) against phage F336. The authors proven that the successful adsorption of the phage F336 to the bacterial surface depends on the hypervariable O-methyl phosphoramidate (MeOPN) modification of capsular polysaccharides (CPS). However, phage resistance has been acquired by loss of MeOPN receptor on the surface of the organism because of cj1421 gene phase variation, which encodesd the MeOPN-GalfNAc transferase [14].

DGRs are genetic elements mediating the receptor–ligand interactions by varying proteins and DNA sequences they encode. The change in phage receptors can be introduced by random mutations and Error-prone DGRs in the genes of bacterial cells encoding cell surface receptors making them incompatible to the phage’s ligand [15].

There are some instances by which phage receptors can be hidden under a physical barrier, such as biofilm, capsule, or another extracellular polymer and confer protective role towards phage attack. In a study, K1 capsules from *E. coli* were shown to interfere with the adsorption of phage T7 to the LPS receptor of this bacterium [12].

### 2.2. Preventing Phage DNA Entry

One of the phage-derived phage defense strategy is superinfection exclusion (Sie). In the case of Sie defense systems phage encoded anti-phage proteins can be utilized by bacterial cells to prevent the translocation of DNA of lytic phage into the cytoplasm of host cells, thereby acquiring protection against virulent phages [16] (Figure 3). These proteins are thought to be associated with membrane components or membrane anchored. The Sie encoding genes are frequently found in prophages (phage genome, which is integrated into the circular bacterial chromosome), signifying that, in many cases, Sie pathways are vital for inter-phage interactions than host–phage interactions. Very limited Sie systems have been fully studied in spite of the fact that various Sie systems have been identified [17,18].

Virulent phages, such as Coliphage T4, have two Sie systems mediated by Sp and imm. These systems prevent the entry of phage DNA into the cytoplasm of bacterial cell, thereby affecting successive infection by other T-even-like phages (Figure 3). The Sp and Imm systems act independently, and their mechanism of action is different from one to the other. Imm protects the direct injection of phage DNA into the cytoplasm of bacterial cells by altering the structural integrity of the translocation site. Imm is suggested to have two transmembrane domains and is expected to be localized to the cytoplasmic membrane, but it does not generate sufficient phage immunity alone. Rather, it must be joined with another membrane protein to be fully functional and attain strong and efficient exclusion [17]. In contrast, the Sp protein, encoded by *gp5*, downregulate the activity of the T4 lysozyme. The activity of the T4 lysozyme encoded by *gp5* has inhibited the membrane protein Sp, thereby likely preventing the peptidoglycan degradation and the consequent translocation of phage DNA [17,19].

### 2.3. Nucleic Acid Interference

Restriction-modification (R-M) systems are universal and tremendously varied in the bacterial primitive immune system. The R–M system has two main components: a methyl transferase (MTase) and a restriction endonuclease (REase). The restriction endonuclease pathway detects short DNA segments, measuring between four to eight base pairs long, and they are chopped into pieces. The mis-recognition and cleavage of the host DNA is protected by the methyl transferase, which is hidden from being recognized by the restriction enzyme [20]. The phage DNA is commonly not methylated and will consequently be degraded upon injection (Figure 4). Naturally competent bacteria such as *Neisseria gonorrhoeae* and others such as *Helicobacter pylori*, *Streptococcus pneumonia* and *Haemophilus influenzae* have abundant R–M systems [21].

There are four R–M systems, so far identified, based on their subunit composition and mechanism of action [22]. Type I and III R-M systems cut and methylate translocated DNA far from the recognition sites. Type II is the most common R–M system, which cut DNA within or near the recognition site. Unlike the other R–M system, type IV systems contain a restriction endonuclease and may or may not have a methylase activity and usually cut the modified DNA [23]. The antiphage potential of a R–M system is directly related to the number of recognition sites found in a phage genome [24,25].

The genome of some virulent phages may be modified by harboring a rare base hydroxymethylcytosine (HMC) in place of the base cytosine. This modification enables T4 phage DNA to be resistant to the classical R–M-based degradation. In co-evolutionary warfare, some bacteria have developed modification-dependent systems (MDSs) that used to attack the modified DNA of phage [26].

### 2.4. Assembly Interference

Phage-inducible chromosomal islands (PICIs) are one of phage resistance mechanism developed by bacteria, which involve the integration of small (∼15 kb) gene sequences and are excised with the help of a specific “helper phage” [27]. The genes encoding for integration factors and excision are localized in the PICI genomes. These genes are switched off by repressors in the absence of helper phages. PICIs are well studied in *Staphylococcus aureus*, where they are specifically located in the pathogenicity islands name as “SaPIs”. In Gram-positive bacteria, PICI expression has been downregulated by a transcription repressor. PICI is excised from the host chromosome by the anti-repressor that has been produced by helper phages. Proteins translated from PICI genome suppress the expression of late helper phage genes and also modify the size of capsid protein to be suitable to accommodate the PICI genome, which ultimately end up with proper packaging of PICI genomes and prevent the formation of helper phage virions [27] (Figure 5).

### 2.5. CRISPR-Cas Systems

CRISPR-Cas is one of the advanced bacterial phage resistance strategies, which has been detected in nearly 50% and 90% of sequenced bacteria and archaea, respectively [28,29,30]. CRISPR systems are acquired immune systems, where long term protection is guaranteed for the second round of infection. In this system, immunological memory is represented by short (30–40 base pairs) “spacers”. Spacers, the hereditary foundations for immunological memory, are generated from phage DNA, and they are flanked by similarly short semi-palindromic repeats. Basically, the *cas*, which we call CRISPR-associated genes, commonly neighboring the CRISPR locus, encode essential proteins needed for gaining new spacers upon infection and for the target-specific removal of the invader. During removal of the invader nucleic acid, RNA-guided Cas nucleases use the crRNAs to identify and cut them via complementary base pairing. The CRISPR systems has been categorized into two classes, six types, and several subtypes, mainly based on the composition of *cas* genes [31], with a wide range of action.

### 2.6. Abortive Infection

Abortive infection (Abi) is a strategy of bacterial cells halt the release of newly produced progeny virions at the expense of the life of infected cell, thereby preventing the uninfected cell from being infected. It is considered a self-sacrificing event, or apoptosis that averts the subsequent infection of the neighboring bacterial community [32]. Each Abi system should contain at least two functional modules: one of these should sense the infection of phage and the other one related to cell death accompanied by the shutdown of metabolism following phage sensing. The Abi system senses intermediate replication of phage genome [33], early, and/or late structural phage proteins [34,35], or phage proteins that are expressed in the cytosol of the cell during replication [36,37]. The Abi system can also sense extensive phage DNA transcription [38,39] or phage-mediated shutoff of host gene expression [40]. Many Abi systems, such as PrrC and Lit, induce cell death because of the inactivation of the host translation system. The Abi gene, *abiZ*, which protects *L. lactis* against phage phi31infection, also induces cell death via cellular membrane damage of infected cells [37]. Premature cell lysis leads to the lysis of the infected cells and results in the release of defective unassembled virions with no potential of infecting other healthy cells [37]. Another Abi system based on *E. coli* data is PifA, which causes abortive infection of phage T7 midway through its replication cycle [34]. An Abi gene was reported to provide protection for *Staphylococcus* species against *Siphoviridae* phages through the phosphorylation of cellular proteins [36]. In general, research findings indicated that most of the Abi system have been identified in *Lactococcus lactis* and *E. coli*, whereby nearly 23 Abi system have been characterized in *L. lactis* [41] (Figure 6).

In recent times, kinase-associated Abi system was identified in *Staphylococcus epidermidis*, protecting them against *Siphoviridae* phages [36]. In this system, following infection, the eukaryotic-like serine/threonine kinase Stk2 was found to phosphorylate several targets of the host machinery, including replication, transcription, translation, and related cellular pathway. This extensive phosphorylation likely interrupts the whole metabolic system to result in abortive infection. Phages counter-attack this system by mutating the pack gene, which affects the activation of Stk2 and its auto phosphorylation [36].

### 2.7. Toxin-Antitoxin Systems

Phages may trigger the bacterial cells to produce toxins, which my attack the phage infection cycle. There are six major bacterial toxin–antitoxin (TA) types, classified based on the toxin neutralization mechanisms, the biomolecular and functional characteristics of the TA and, the TA count (some bacteria containing lots of TA gene pairs, such as *E. coli* K-12, which has more than 35 TA pairs) [42]. This wide genetic diversity indicated that the TA system is responsible for many functions: apart from phage resistance, they have been involved in plasmid maintenance, stress responses, and persisting cell production. The replication pathway of phages can be directly affected by some of the TA system. For example, the MazF/MazE TA system of *E. coli* can inhibit the infection process of T4 phage by inducing MazF’s ribonuclease activity, which ultimately leads to complete cessation of the infection cycle [43]. Table 1 summarizes some of the different anti-phage defense mechanisms across various species of bacteria.

### 2.8. Bacterial Retrons

Retrons are bacterial genomic materials composed of a non-coding RNA (ncRNA) and reverse transcriptase (RT). The ncRNA served as a template for RT, producing a chimeric DNA/RNA molecule in which the DNA and RNA are covalently connected [56]. Retrons have been named in accordance with a convention, encompassing the first genus letters followed by species names, as well as their reverse-transcribed DNA lengths (for example, *E. coli* retron, Ec48, has 48 nt long reverse-transcribed DNAs [57]). It has been three decades since retrons were discovered, but little is known about their function. However, the anti-phage activity of retrons has been currently reported. Millman and coworkers reported the role of retrons as an anti-phage defense strategy. This defense system is consisted of three principal units: the ncRNA, RT, and an effector protein. According to the result obtained from this study, the phage proteins inhibit *E. coli* RecBCD, causing the retron (Ec48) to activate and kill the cell via Abi [58]. The authors suggested that retrons serve as second defense line if the first defense line has collapsed (Figure 7).

### 2.9. Bacterial Secondary Metabolites (Chemical Agents)

Antiphage defense systems discussed so far have largely been mediated by RNA or proteins complexes acting on individual cells. However, bacterial secondary metabolites, such as gibberellins, toxins, alkaloids, and biopolymers, can interfere with the replication cycle of phages via different mechanisms and defending them from phage attack [59]. Maxwell and coworkers introduced a chemical anti-phage defense mechanism that predominantly occurred in soil fungus, *Streptomyces* spp. They reported that *Streptomyces* spp. synthesized two vital bioactive metabolites (doxorubicin and daunorubicin) that bind (intercalate) into the DNA of phage and prevent the replication cycle. Interestingly, the bacterial growth pattern was not affected by these molecules. It has been proposed that daunorubicin exerted its action at an early stage in the phage replication cycle, after the translocation of the DNA, but ahead of replication. Doxorubicin forms free radicals (e.g., OH^•^), which can damage DNA and cause DNA oxidation (Figure 8). The authors also reported that these metabolites can disperse into bacteria cells and safeguard them from infection [60]. Idarubicin and pirubicin are other DNA intercalating agents, which are produced by *Streptomyces* spp. These metabolites are thought to affect the circularization process of the phage linear DNA or to affect proteins that are involved in the transcription and translation process [59,61].

## 3. Phage Counteracting Mechanisms

Almost all living organisms, including archaea, fungi, and bacteria, are frequently infected with viruses and have developed miscellaneous means of resistance. On the other side, viruses develop counter-defense strategies. Viruses counterattack the host defense strategies in different ways. Some of the strategies are discussed below.

### 3.1. Access to Host Receptors

Phages counteract the bacterial resistance associated with their receptors in many different mechanisms. Adaptation of phages to a new bacterial receptor or unhiding of bacterial cell surface receptors using phage derived enzymes are the two major counterstrategies employed by phages [62,63] (Figure 9). The RBPs of tailed phages, which taxonomically belong to the family *Siphoviridae*, can be modified to generate a new receptor towards the target hosts. Recent findings indicated that the phage λ evolves to target a new receptor, OmpF, which previously used the LamB receptor on the surface of the *E. coli* B. The expression of the equivalent receptor, LamB, is suppressed via mutation [62]

### 3.2. Anti-Restriction—Modification Systems

Bacterial cell typically used a restriction endonuclease (REase) enzyme, which cleaves the phage DNA at specific recognition site(s). In response to this defense feature, phages developed a broad range of passive and active anti-restriction strategies [22,64] (Figure 10).

#### 3.2.1. Active Evasion Mechanisms

The *Myoviridae* coliphage P1 encodes two inti-restriction proteins, DarA and DarB, that are co-translocated into host cells, along with their genome, thereby hiding the type I R-M recognition sites and protecting phage DNA degradation [65]. In addition, Ocr protein of coliphage T7 mimics the structure of phosphate backbone of the cellular DNA and directly interacts with EcoKI (both the REase and MTase domain of this type I R–M enzyme), thus affecting the action of this system [66,67].

#### 3.2.2. Passive Mechanisms of Phage Evasion

In the case of passive mechanisms of phage evasion, MTase modifies the double-stranded DNA of phages rapidly within a host comprising a R–M system before it is recognized by the host REase. Thus, the invading phage DNA will be protected from degradation by R–M systems. Hence, the modified genome of phage can replicate safely in the R–M-consisting host cell and can also infect and replicate in other cells which express identical R–M system. Nevertheless, as the R–M system is specific to specific host, the same DNA of a phage can be detected as foreign in a cell consisting of a different R–M system, and it will, therefore, be degraded by a different REase. In yet another twist, some REases of bacterial cell even can detect and degrade modified DNA [68].

### 3.3. Escaping Abortive-Infection Mechanisms

Certain phages have developed anti-Abi mechanisms to undergo protected replication in the targeted host cell [69]. *Lactococcus* phages are the best example for this counterstrategy. Mutations of one or more specific genes of the *Lactococcus* spp. can enable the phages to escape the Abi systems [70]. Phages from *Lactococcus* spp., for instance, can bypass the AbiQ mechanism by mutating genes involved in nucleotide metabolism. A phage can encode a molecule (such as Dmd in coliphage T4) that can replace a bacterial antitoxin, thus inhibiting the activity of the bacterial toxin and protecting the cell from death [71]. It has been discovered that the *Pectobacterium atrosepticum* phage, phiTE, produces pseudo-anti-toxin RNA or takes over the antitoxin, ToxI, during its infection to deactivate the ToxN toxin [72].

### 3.4. Evading CRISPR–Cas Systems

#### 3.4.1. Evasion by Mutation

An individual nucleotide substitution can enable phages to evade the CRISPR system in the protospacer site or in the conserved region adjacent to the protospacer motif [73]. In contrast to phages with multiple mismatches in distal PAM protospacer sites, those with substitutions near the protospacer-adjacent motif will evade CRISPR targeting [74].

#### 3.4.2. Evasion by Anti-CRISPR Genes

Current research findings indicated that phages encode anti-CRISPR genes, which are active towards the CRISPR of bacterial defense systems, as it is recognized in *Pseudomonas aeruginosa* temperate phages [75]. Some phages encode several anti-CRISPR proteins (Acrs) that interfere with the function of different variants of the CRISPR–Cas system [76].

Some examples of the phage’s counterstrategies towards different phage resistance mechanisms that we discussed so far are summarized in Table 2.

## 4. Conclusions and Future Directions

In this review, we have reviewed a variety of phage resistance mechanisms. These antiviral defense systems involved several biomolecules, proteins, enzymes, cellular structures, and inter-cellular interactions that protect bacterial cells from lysis and death. The bacterial antiphage defense strategies, which are described in this review, reflect the tremendous diversity of phages, and thus some other resistance mechanisms could be discovered.

The study of phage resistance should be scaled up beyond the discovery of the baseline mechanism towards the detailed understanding on the molecular root behind these antiviral systems. In this context, advancement in phage biology will certainly be required to fully understand the mechanism of phage resistance. Moreover, most studied antiphage systems were associated with the phages whose genomic matter is double-stranded-DNA. Resistance mechanisms linked to single-stranded DNA and RNA, or double-stranded RNA genomes, as well as other genetically unique phages, are waiting to be investigated. In addition, lysogenic phage mediated resistance mechanisms are less understood, and therefore this gap needs to be filled.

The above-mentioned phage resistance strategies are often investigated in a laboratory-controlled setting, in a single replica, and using a single host–phage model. Nevertheless, bacterial pathogens usually comprised multi-lateral antiphage systems. The coexistence of these systems in one host cell has seldom been studied, and the outcome of such interactions on phage evolution and their community is often overlooked. Likewise, the effectiveness of phage resistance strategies towards some families of phages (non *Siphoviridae*) are not continually evaluated.

As bacteria and phages have ancient co-evolutionary history, phages can profoundly generate a counter-resistance, via several mechanisms and with small fitness cost. One phage may also possess multiple phage resistance systems that may generate strong defenses over individual systems, allowing specific clonal inhabitants to stay in phage-containing ecosystems. The phage counteracting strategies enable the phages to undergo their replication safely and to produce viable progeny virions for the next generation. However, there are many unexplored mechanisms, cascades of reactions, molecular interactions, and involvement of newly produced biomolecules that need additional in-depth investigation for better understanding of phage science. Hence, the limitation of this review is that we have presented limited information regarding the mechanisms of action of some phage resistance and counterstrategies that have not yet been fully explored.

## Figures and Tables

**Figure 1 antibiotics-12-00381-f001:**
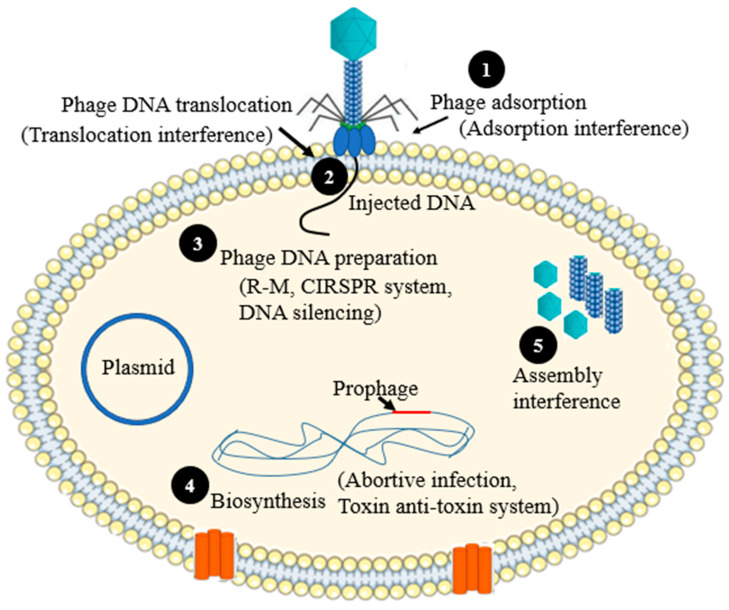
Schematic illustration of the mechanisms of phage resistance in relation to the life cycle of phages. (**1**) Adsorption (phage can use one or multiple receptor binding protein (RBPs) to adhere on the host surface. Bacterial strains can interfere this phase by different mechanisms, such as masking of the receptor(s), changing the structural organization the receptor(s), etc.); (**2**) Translocation (direct injection of phage DNA into host cytoplasm). Some bacterial strains can interfere with the injection of phage nucleic acid by the host encoded proteins; (**3**) Phage DNA preparation in the cytoplasm. Either modified or unmodified phage DNA can be degraded with the host encoded proteins following its translocation into the host cytoplasm; (**4**) Replication (biosynthesis). Some bacterial strains produce antiviral proteins that can interfere with the phage biosynthesis pathway by interacting with one or more structural or nonstructural proteins of the infected phage, or they may use phage-encoded proteins to protect themselves from phage attack(s); (**5**) Assembly and packaging (some bacterial strains may express proteins, which can interfere the process of assembly or packaging of newly produced virions).

**Figure 2 antibiotics-12-00381-f002:**
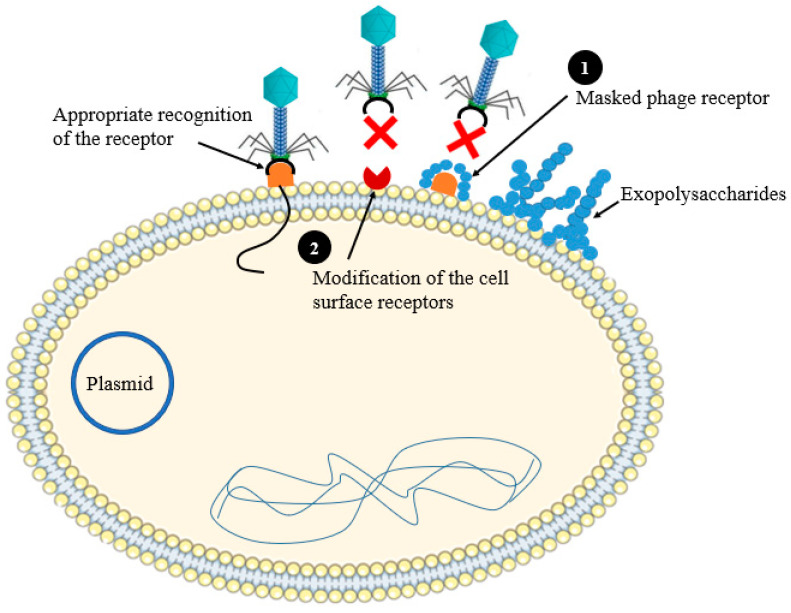
Schematic illustration of the phage resistances mechanism at the point of phage adsorption. (**1**) Masking the phage receptor with exopolysaccharides; (**2**) Modification of the bacterial cell surface receptors.

**Figure 3 antibiotics-12-00381-f003:**
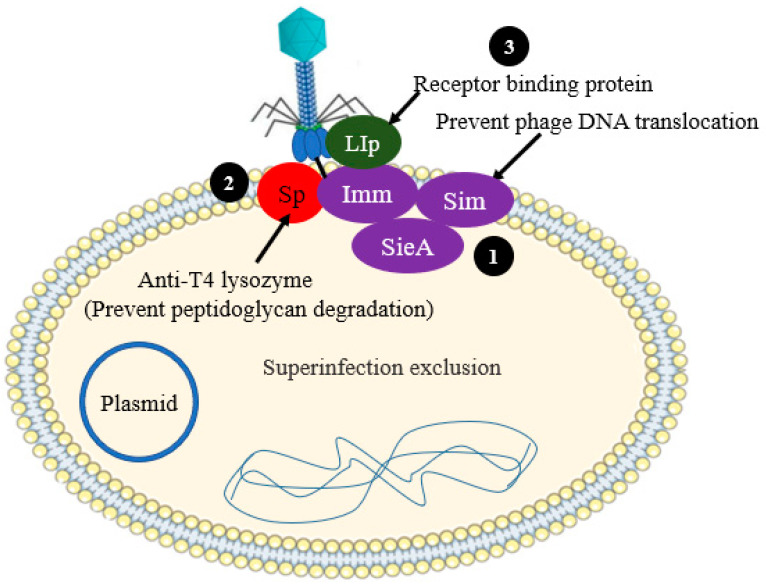
Schematic illustration of the superinfection exclusion systems. (**1**) Phage proteins prevent phage DNA translocation—Imm (T4 phage), SieA (P22 phage), Sim (P1 phage). (**2**) Prevent phage peptidoglycan layer degradation by inhibiting T4 lysozyme activity (Sp of T4 phage). (**3**) Phage protein bind to the receptor (e.g., LIp prevents the entry of T5 phage).

**Figure 4 antibiotics-12-00381-f004:**
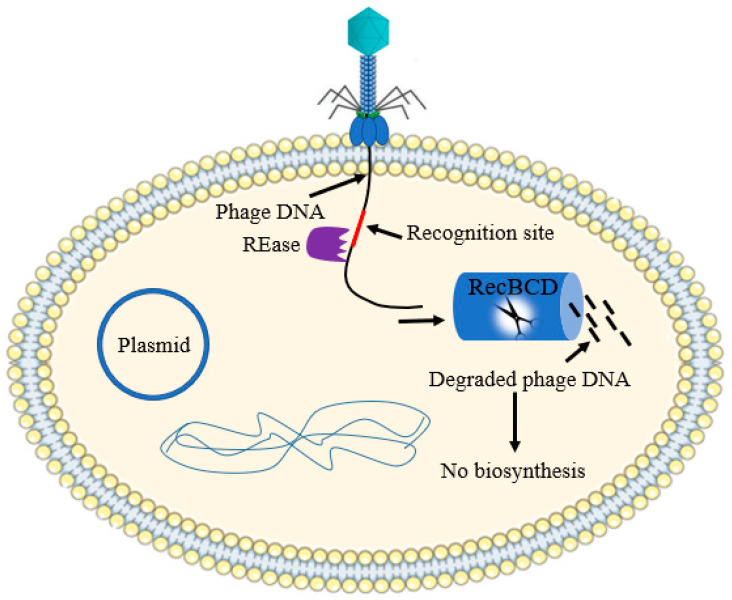
Schematic illustration of bacterial R–M systems. R–M systems efficiently degrade phage DNA once introduced into the cytosol of the host cell. Fragmented translocated DNA is recognized as non-self (lacking chi sequences) and is subjected for further degradation by RecBCD enzyme.

**Figure 5 antibiotics-12-00381-f005:**
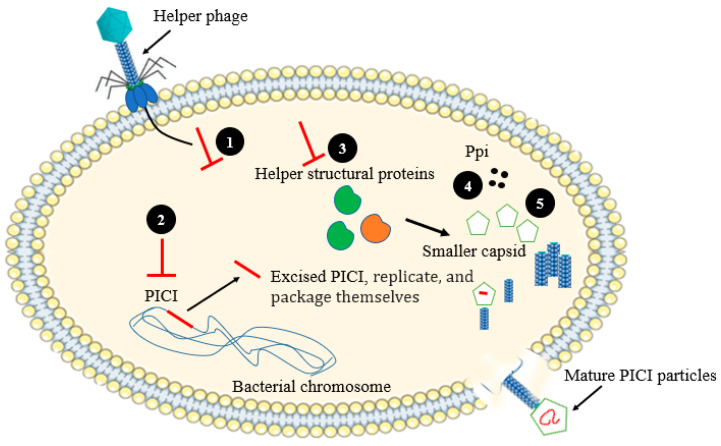
Schematic diagram showing the mechanisms of PICI-based assembly interference. (**1**) Translocation of helper phage genome may activate host protein, which interfere the activation of helper phage early genes that are involving in the early replication cycle; (**2**) Excised PICI from the PICI harboring genome; (**3**) Interference with the activation of helper phage late genes; (**4**) Helper phage packaging interference (Ppi) proteins; (**5**) Small capsid avoids the packaging of larger-size helper phage genome.

**Figure 6 antibiotics-12-00381-f006:**
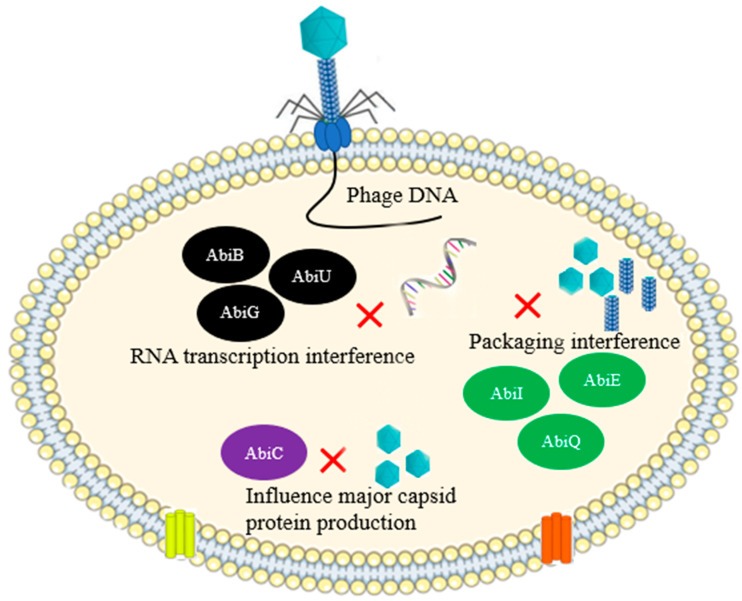
Abi pathway of *L. lactis*. Three proteins AbiU, AbiG and AbiB interfere with RNA transcription. AbiC limit major capsid protein production. AbiQ, AbiI, and AbiE interfere with the packaging of phage DNA.

**Figure 7 antibiotics-12-00381-f007:**
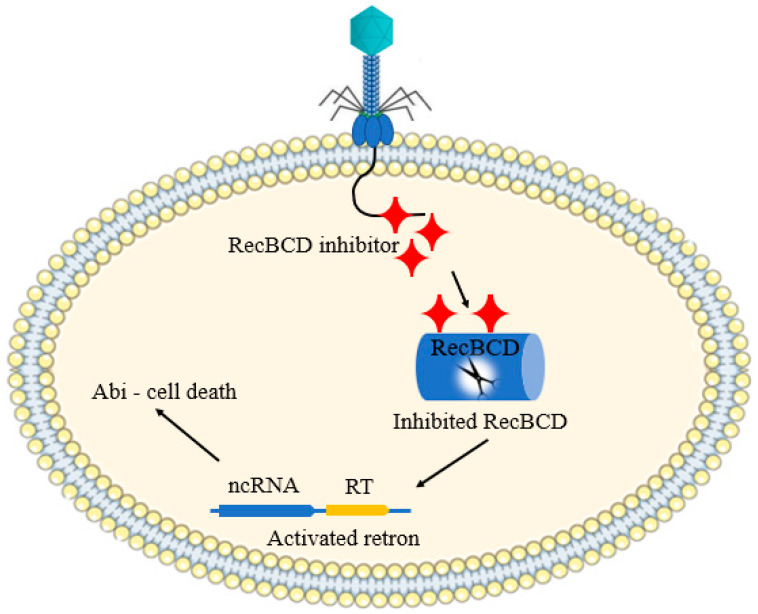
Schematic illustration of bacterial retron-based anti-phage defense mechanism.

**Figure 8 antibiotics-12-00381-f008:**
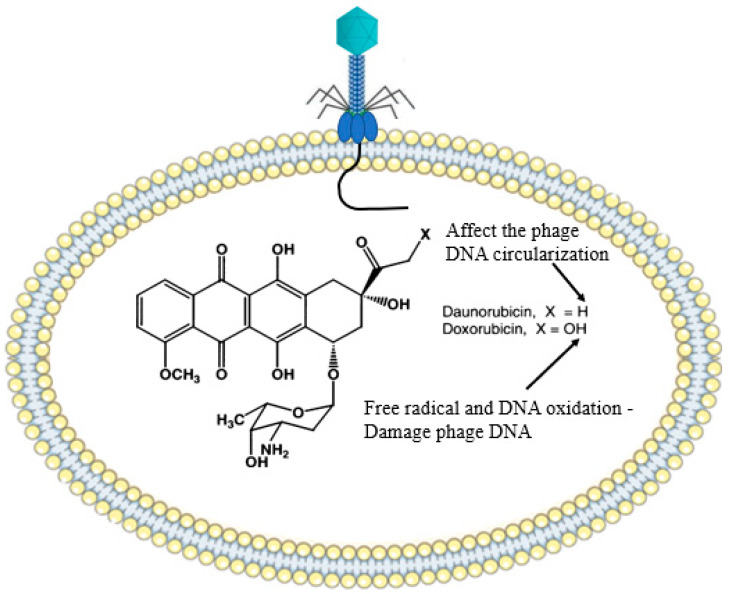
Schematic diagram elucidating the mechanism of action of bacterial secondary metabolites in relation to the phage infection cycle. Daunorubicin mainly affect the phage DNA circularization and exposing them for restriction enzyme digestion. Doxorubicin attacks the phage DNA with its free radical (OH.), and the subsequent oxidation will result in full degradation.

**Figure 9 antibiotics-12-00381-f009:**
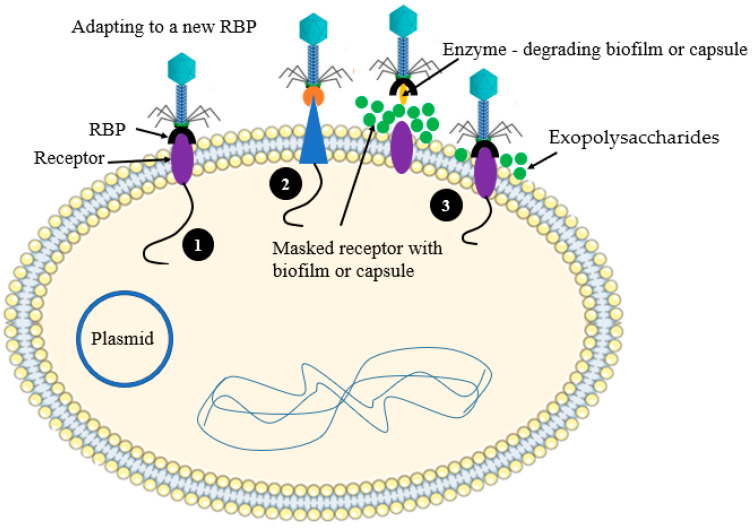
Schematic illustration of the mechanism of phage accesses the receptor of phage on the surface of host cell. (**1**) Adapting to a new RBP; (**2**) Adapting to a modified RBP; (**3**) Access the host cell receptor by enzymatic degradation of the exopolysaccharides.

**Figure 10 antibiotics-12-00381-f010:**
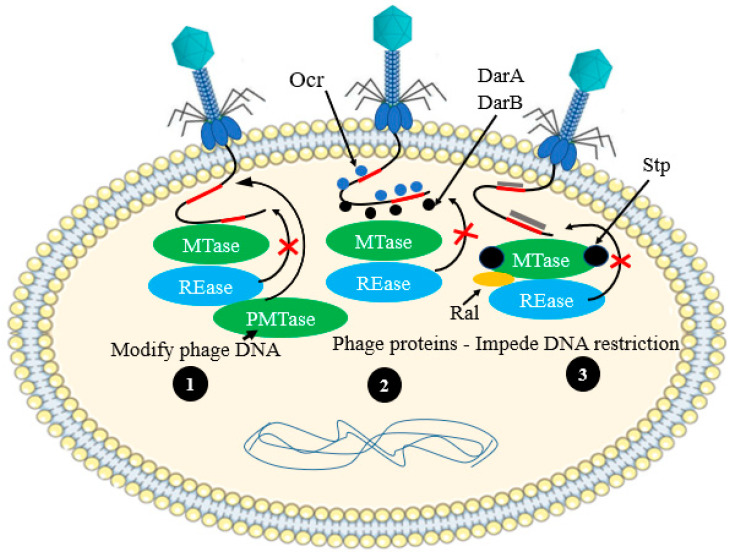
Schematic representation of phage anti-restriction–modification systems. (**1**) Host MTase can modify the genome of phages, which enable the agent to undergo safe replication in the cytoplasm of the host cell. On the other hand, MTase can be encoded by the genome of phage and expressed its own MTase (PMTase) during infection; (**2**) Phages (e.g., phage P1) can simultaneously inject proteins, such as DarA and DarB with its DNA to bind to the DNA of phage and hidden the restriction sites. The target phage DNA can be mimicked by a phage protein (e.g., Ocr of phage T7) binded to both REase and MTase, and it seizes the restriction enzyme; (**3**) The activity of the MTase can be activated by phage derived proteins, such as Ral of phage λ and thereby quicken the phage DNA protection. The phage T4 peptides, such as Stp, can also impede restriction by perturbation of the MTase–REase system.

**Table 1 antibiotics-12-00381-t001:** Some examples of anti-phage defense strategies across different bacterial species.

Specific Systems	Bacteria	Phage Resistance Mechanism (Anti-Phage Defense Strategies)	References
ToxN, RNase activity, destroying both host and phage transcripts	*Pectobacterium atrosepticum*	TA systems	[44]
MazF/MazE TA system	*E. coli*	TA systems	[43]
Phage-inducible chromosomal islands (PICIs)	*Staphylococcus aureus*	Assembly Interference	[45]
Ppi protein prevent phage packaging process	*Staphylococcus* spp.	Assembly interference	[46]
abiK system	*L. lactis*	Abi	[47]
AbiZ (100-fold reduction of the burst size of phage Φ31)	*L. lactis*	Abi	[37]
Stk2 Abi System (kinase-mediated Abi mechanism)	*Staphylococcus epidermidis*	Abi	[36]
Inhibiting the protein translation system using the peptide Lit and the anticodon nuclease (PrrC)	*E. coli*	Abi	[48]
Stp protein of phage T4 affects the interaction of PrrC and EcoprrI, freeing triggered PrrC protein and leading to abortion	*E. coli*	Abi	[33]
“inverted” RM systems	*Streptomyces coelicolor* A2(3)	R–M systems	[49]
Type IV pili	*Pseudomonas aeruginosa*	Preventing phage adsorption	[50]
Lipopolysaccharide (LPS)	*E. coli* K1	Preventing phage adsorption	[12]
Mediated by imm and sp	*E. coli*	Sie systems	[17]
Sie2009	*L. lactis* and lactococcal prophages	Sie systems	[51,52]
Using the signal peptide lipoprotein prophage TP-J34 (LTP)	Prophage of *Streptococcus thermophilus*	Sie systems	[53]
Using MDS enzymes (DpnI for Streptococcus pneumoniae, McrBC, McrA, and Mrr for *E. coli*)	*Streptococcus pneumoniae*, *E. coli*	Modification-dependent systems (MDSs), which detect the modified phage DNA	[26,54]
RNA-guided DNA silencing and DNA-guided DNA silencing	It is available in some bacterial spp.	Argonautes (pAgos)	[55]

**Table 2 antibiotics-12-00381-t002:** Some examples of phage counterstrategies.

Specific System	Anti-Phage Resistance Strategies	Example of Phages	References
Absence of endonuclease recognition sites (Lack of *Sau* 3A regions in its dsDNA)	Anti-R-M system	*Staphylococcus* phage K	[77]
Phage DNA modification comprised the rare base hydroxymethylcytosine (HMC) in place of the cytosine	Anti-R-M system	Phage T4	[26]
Ocr protein prevent restriction activity	Anti-R-M system	Coliphage T7	[67]
Using anti-restriction protein	Anti-R-M system	Phage P1	[65]
Using protein RIIA and RIIB	Anti-Abi mechanism	Phage T4	[33]
The toxin effect of LsoA and RnlA neutralized by Dmd during replication of phage	Anti-Abi system	Coliphage T4	[71]
Polysaccharide-degradation using hydrolases and lyases	Accessing the receptors	Depolymerase producing phages	[78]
Anti-CRISPR proteins	Interfere with CRISPR–Cas system	*P. aeruginosa* prophages	[75]
Protospacer mutation	Interfere with CRISPR–Cas system	*Streptococcus thermophilus* phages	[73]
Mutations in the RBP-encoding gene mutation	New receptors adaptation	Coliphages T7 and ϕX174	[79]
Tail protein modification	New receptors adaptation	*Pseudomonas fluorescens* phage ϕ2, *L*. *lactis* phage LL-H	[80,81]

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
