# Peer review of "The Battle between Bacteria and Bacteriophages: A Conundrum to Their Immune System"

_antibiotics, 2023, doi:10.3390/antibiotics12020381_

Round 1

Reviewer 1 Report

The manuscript is well-written and has substantial scientific merit.

The authors need to include the limitations of the review.

However, there are some minor spelling/grammatical errors that need to be corrected by the authors.

I will suggest that the authors go over the manuscript once more to correct these minor errors.

Author Response

Manuscript ID: Antibiotics- 2200858

Response to Reviewer #1 Comments

 Dear Editor,

Thank you so much for communicating to us the reviewer’s comments and suggestions. We truly appreciate your input and reviewers comment concerning our manuscript entitled “The battle between bacteria and bacteriophages: a conundrum to their immune system.  Your   reviewer’s comments   were   very helpful to improve our paper.   We   have   carefully   addressed all comments and suggestions. Our responses are presented below. All the changes are marked in yellow color in the revised manuscript.

We look forward to your decision. We will be more than happy to make any further changes that might arise. Thank you again for your cooperation and the consideration of our manuscript.

Sincerely,

Reviewer comments and response

Comments and Suggestions for Authors

The manuscript is well-written and has substantial scientific merit.

Response: Thank you so much.

1.The authors need to include the limitations of the review.

Response: Thanks. The limitation has been included. (See page 17/18, line 459-460).

2.However, there are some minor spelling/grammatical errors that need to be corrected by the authors.

Response: Thanks. Corrections have been made throughout the manuscript.

I will suggest that the authors go over the manuscript once more to correct these minor errors.

Reviewer 2 Report

This manuscript reviews the range of biochemical responses elicited in bacteria upon attack by bacteriophages and the corresponding biochemical responses of the bacteriophages used to evade bacterial response mechanisms and immunity. The review is generally clearly written and appropriate and useful schematics accompany the text and provide summaries of the key biological defence mechanisms within the bacteria and the bacteriophage responses. An appropriate range of key literature is summarised. Whilst the topic has been covered in reviews before and the review is comparatively broad in scope, I feel that it nevertheless provides a useful summation and interpretation of the latest literature in this field. There are some points that the authors could address and these are detailed below (assume all points require changes to the manuscript).

1.       Page 1 line 25 Change “. . . genomic mater . . .” to “. . . genomic matter . . .” here and throughout the manuscript.

2.       Page 3 lines 102-103 “. . . hidden receptors with . . . slime layer, capsule) . . .” This description needs some more detail and examples. What is the role of a capsule in this context?

3.       Page 4 lines 137-138 “. . . signifying that in . . . host-phage interactions.” Your meaning is not clear here. Reword this statemen to clarify your meaning.

4.       Page 5 line 147 Consistent and meaningful definitions need to be provided in the text for sp and imm. Also, you need to ensure your capitalisation of Sp is consistent (it differs across Figure 3 and the main text).

5.       Page 6 lines 174-176 “Unlike the other . . . the modified DNA.” This does not appear to be always so. Lepikhov et al. reported methyltransferase activity in a type IV restriction-modification system (Lepikhov et al. Nucleic Acids Res. 2001 29(22) 4691-4698).

6.       Page 8 lines 227-228 “. . . where nearly 23 . . . (Figure 6).” “Abi systems” are ill-defined in the text and more detail is needed on what constitutes an “Abi system”.

7.       Page 10 lines 281-282 “. . . two vital bioactive . . . the replication cycle.” Doxorubicin and daunorubicin function by intercalation. Interpolation has an entirely different meaning in genetics. Correct this text accordingly.

8.       Page 10 lines 285-286 “Doxorubicin forms free . . . oxidation (Figure 8).” This release of free radicals presumably also runs the risk of damaging the bacterial DNA in addition to the bacteriophage DNA. Has there been any attention paid in the scientific literature to this unfortunate lack of specificity?

9.       Page 13 line 367 “. . . PAM . . .” This abbreviation should be defined at first use in the manuscript text.

10.   Table 2. The column formatting needs to be improved to avoid word breaks.

11.   Page 15 lines 405-406 “. . . the union of . . . the other . . .” It is not clear what you mean by “the union of one phage resistance system”. Reword to clarify your meaning.

12.   The manuscript contains a number of grammatical errors that the authors should check and correct (too many to list individually).

Author Response

Manuscript ID: Antibiotics - 2200858

Response to Reviewer #2 Comments

 Dear Editor,

Thank you so much for communicating to us the reviewer’s comments and suggestions. We truly appreciate your input and reviewers comment concerning our manuscript entitled “The battle between bacteria and bacteriophages: a conundrum to their immune system.  Your   reviewer’s comments   were   very helpful to improve our paper.   We   have   carefully   addressed all comments and suggestions. Our responses are presented below. All the changes are marked in yellow color in the revised manuscript.

We look forward to your decision. We will be more than happy to make any further changes that might arise. Thank you again for your cooperation and the consideration of our manuscript.

Comments and Suggestions for Authors

This manuscript reviews the range of biochemical responses elicited in bacteria upon attack by bacteriophages and the corresponding biochemical responses of the bacteriophages used to evade bacterial response mechanisms and immunity. The review is generally clearly written and appropriate and useful schematics accompany the text and provide summaries of the key biological defense mechanisms within the bacteria and the bacteriophage responses. An appropriate range of key literature is summarized. Whilst the topic has been covered in reviews before and the review is comparatively broad in scope, I feel that it nevertheless provides a useful summation and interpretation of the latest literature in this field. There are some points that the authors could address, and these are detailed below (assume all points require changes to the manuscript).

Reviewer comments and response    

  1. Page 1 line 25 Change “. . . genomic mater . . .” to “. . . genomic matter . . .” here and throughout the manuscript.

Response: Thanks. This observation is correct and corrected accordingly. (See page 1, line 29).

  1. Page 3 lines 102-103 “. . . hidden receptors with . . . slime layer, capsule) . . .” This description needs some more detail and examples. What is the role of a capsule in this context?

Response: Thanks. This observation is correct and corrected accordingly. (See page 3, line 107 - 112).

  1. Page 4 lines 137-138 “. . . signifying that in . . . host-phage interactions.” Your meaning is not clear here. Reword this statemen to clarify your meaning.

Response: Thanks. It has been corrected. (See page 4, line 149 -150).

  1. Page 5 line 147 Consistent and meaningful definitions need to be provided in the text for sp and imm. Also, you need to ensure your capitalization of Sp is consistent (it differs across Figure 3 and the main text).

Response: Thanks. It has been corrected. (See page 6, line 152 -171).

  1. Page 6 lines 174-176 “Unlike the other . . . the modified DNA.” This does not appear to be always so. Lepikhov et al. reported methyltransferase activity in a type IV restriction-modification system (Lepikhov et al. Nucleic Acids Res. 2001 29(22) 4691-4698).

Response: We agree with this comment and the required information has been added. Corrected. (See page 7, line 188 -189).

  1. Page 8 lines 227-228 “. . . where nearly 23 . . . (Figure 6).” “Abi systems” are ill-defined in the text and more detail is needed on what constitutes an “Abi system”.

Response: We thank the reviewer for this comment. It has been corrected. (See page 9, line 142 -167).

  1. Page 10 lines 281-282 “. . . two vital bioactive . . . the replication cycle.” Doxorubicin and daunorubicin function by intercalation. Interpolation has an entirely different meaning in genetics. Correct this text accordingly.

Response: Thanks. It has been corrected. (See page 12, line 324).

  1. Page 10 lines 285-286 “Doxorubicin forms free . . . oxidation (Figure 8).” This release of free radicals presumably also runs the risk of damaging the bacterial DNA in addition to the bacteriophage DNA. Has there been any attention paid in the scientific literature to this unfortunate lack of specificity?

Response: Thanks for your information unfortunately we couldn’t find any information about the point that you have mentioned. However, from our point of view such a phenomenon may be associated with the Abi system which result into the death of bacterial cells following the DNA damage. Such kind of death usually protect the uninfected bacterial cells since the virions which are going to be released from the dead bacterium wound be immature or defective or non-infectious.

  1. Page 13 line 367 “. . . PAM . . .” This abbreviation should be defined at first use in the manuscript text.

Response: Thanks. It has been corrected. (See page 15, line 414).

  1. Table 2. The column formatting needs to be improved to avoid word breaks.

Response: Thanks. This is usually managed by the publisher.

  1. Page 15 lines 405-406 “. . . the union of . . . the other . . .” It is not clear what you mean by “the union of one phage resistance system”. Reword to clarify your meaning.

Response: Thanks. It has been corrected. (See page 17, line 152 -153).

  1. The manuscript contains several grammatical errors that the authors should check and correct (too many to list individually).

Response: Thanks. It has been corrected.

Reviewer 3 Report

The manuscript: ‘The battle between bacteria and bacteriophages: a conundrum to their immune system’ describes broadly bacterial anti-phage defence mechanisms and phages responses to them. It also suggests that the discovered systems are mostly against DNA phages and thus just the beginning. The manuscript is well structured and covers most of the current knowledge. However, anti-phage systems against lysogenic phages such as DISARM and BREX molecular systems, could be mentioned. All the figures are informative but in many cases the legends can be improved. Additionally, there are quite many spelling or language mistakes which I try to address below.

Minor comments:

These are suggestions how I would modify the text.

Row 25: Most studied antiphage systems were associated à Most studied antiphage systems are associated

Row 31: retrons and secondary metabolites-based replication interference and so on. à retrons and secondary metabolites-based replication interference.

Row 33: These mechanisms allowed phages to undergo their à These mechanisms allow phages to undergo their

Row 58: phage-insensitive hosts protecting bacterial lineages, while counter-resistant phages threatening such strains. à phage-resistant hosts protecting bacterial lineages, while counter-resistant phages threatening such strains.

Row 129: Modification or expression of new RBP à This is a bit misleading sentence because it mentions phage protein where previously the figure text is about host defence it needs to be modified

Row 138: Very limited Sie systems have been characterized even if various Sie systems have been identified

Please modify above sentence, it is not formulated well

Row 143: Prevent phage peptidoglycan layer degradation by inhibiting T4 lysozyme activity (Sp of T4 phage);

This is not continuing well the list. Maybe add or in the beginning or start with determining what prevents.

Row 151: Imm protects not Imm protect

Row 166: which is instead of which

Row 186: recognized à is recognized

Entire figure 5 describtion on rows 203-206 should be modified.

Row 212: Spacer à Spacers

Row 294: manly à mainly

Row 318: Why these anti-restriction strategies are not mentioned here instead of referring previous parts of the text?

Row 329: Remove with

Row 331: This sentence does no make sense, please modify: f the MTase modifies double-stranded DNA of phages within a host consisting of an 331 R–M system rapidly before the REase recognizes it, and the invading phage DNA will be 332 protected from R-M based enzymatic degradation

Row 383: Please remove this: has been ultimately

After these modifications I recommend publishing of the manuscript.

Author Response

Manuscript ID: Antibiotics- 2200858

Response to Reviewer #3 Comments

 Dear Editor,

Thank you so much for communicating to us the reviewer’s comments and suggestions. We truly appreciate your input and reviewers comment concerning our manuscript entitled “The battle between bacteria and bacteriophages: a conundrum to their immune system.  Your   reviewer’s comments   were   very helpful in improving our paper.   We   have   carefully   addressed all comments and suggestions. Our responses are presented below. All the changes are marked in yellow color in the revised manuscript.

We look forward to your decision. We will be more than happy to make any further changes that might arise. Thank you again for your cooperation and the consideration of our manuscript.

Sincerely,

Reviewer comments and response

  1. Row 25: Most studied antiphage systems were associated à Most studied antiphage systems are associated

Response: Thanks. It has been corrected. (See page 1, line 28).

  1. Row 31: retrons and secondary metabolites-based replication interference and so on. à retrons and secondary metabolites-based replication interference.

Response: Thanks. It has been corrected. (See page 1, line 34).

  1. Row 33: These mechanisms allowed phages to undergo their à These mechanisms allow phages to undergo their

Response: Thanks. It has been corrected. (See page 1, line 36).

  1. Row 58: phage-insensitive hosts protecting bacterial lineages, while counter-resistant phages threatening such strains. à phage-resistant hosts protecting bacterial lineages, while counter-resistant phages threatening such strains.

Response: Thanks. It has been corrected. (See page 2, line 61).

  1. Row 129: Modification or expression of new RBP à This is a bit misleading sentence because it mentions phage protein where previously the figure text is about host defence it needs to be modified

Response: Thanks. It has been corrected. (See page 4, line 139 -140).

  1. Row 138: Very limited Sie systems have been characterized even if various Sie systems have been identified. Please modify above sentence, it is not formulated well

Response: Thanks. It has been corrected. (See page 5, line 149 -151).

  1. Row 143: Prevent phage peptidoglycan layer degradation by inhibiting T4 lysozyme activity (Sp of T4 phage);

This is not continuing well the list. Maybe add or in the beginning or start with determining what prevents.

Response: Thanks. It has been corrected. (See page 6, line 157).

  1. Row 151: Imm protects not Imm protect

Response: Thanks. It has been corrected. (See page 6, line 164).

  1. Row 186: recognized à is recognized

Response: Thanks. It has been corrected. (See page 7, line 200).

  1. Entire figure 5 description on rows 203-206 should be modified.

Response: Thanks. It has been corrected. (See page 8, line 218 - 223).

  1. Row 212: Spacer à Spacers

Response: Thanks. It has been corrected. (See page 8, line 229/232).

  1. Row 294: manly à mainly

Response: This observation is correct and corrected accordingly. (Page 13, line 336).

  1. Row 318: Why these anti-restriction strategies are not mentioned here instead of referring previous parts of the text?

Response: Thanks. The restriction system that you have mentioned have been discussed starting from page 14, line 367.

  1. Row 329: Remove with

Response: This observation is correct and corrected accordingly. (See page 14, line 372).

  1. Row 331: This sentence does no make sense, please modify: f the MTase modifies double-stranded DNA of phages within a host consisting of an 331 R–M system rapidly before the REase recognizes it, and the invading phage DNA will be 332 protected from R-M based enzymatic degradation

Response: We thank the reviewer for this comment. Corrected. (See page 14, line 374-379).

  1. Row 383: Please remove this: has been ultimately

Response: This observation is correct and corrected accordingly. (See page 17, line 430).
